# Fabrication of an Anti-Icing Aluminum Alloy Surface by Combining Wet Etching and Laser Machining

**Annan Xia** [†]**, Lei He** [†]**, Shihang Qie, Jingchen Zhang, Hanlong Li, Ning He and Xiuqing Hao** *

College of Mechanical and Electrical Engineering, Nanjing University of Aeronautics & Astronautics, Nanjing 210016, China; anxia@nuaa.edu.cn (A.X.); meelhe@nuaa.edu.cn (L.H.); ksh111@nuaa.edu.cn (S.Q.); zjc_nuaa@nuaa.edu.cn (J.Z.); einzbernvonzz@126.com (H.L.); drnhe@nuaa.edu.cn (N.H.)
* Correspondence: xqhao@nuaa.edu.cn
† These authors contributed equally to this work.

**Abstract:** The phenomenon of icing on the surface of the fuselage while aircraft pass through clouds has an impact on flight safety. This aircraft icing may adversely affect metrological conditions and cause aerodynamic mechanical effects, resulting in a threat to flight safety. This research aims to fabricate an anti-icing surface on a 2524 aluminum alloy material by combining laser machining and wet etching. The microstructure surfaces were obtained by laser, the nanostructured surfaces were obtained by wet etching, and the hierarchical structures were prepared through a combination of these two processes. The contact angle, icing delay performance, icing adhesion, and dynamic water repellency of three kinds of textured surfaces and smooth surface were tested and compared comprehensively through experiments. The experimental findings have shown that the hierarchical surface has the best superhydrophobic properties, and the freezing time of droplets on the hierarchical surface can be extended 10 times. In addition, the ice adhesion on the surface of the hierarchical structure decreased by approximately 75% compared to the untreated samples. The surface of the hierarchical structure showed the best dynamic water repellency. The proposed hybrid laser machining–wet etching fabricating method has the potential to avoid aircraft icing.

**Keywords:** aluminum alloy; micro/nanostructure; anti-icing performance; superhydrophobic surface



## 1. Introduction

Aircraft icing refers to the accumulation of ice on the windward side of an aircraft [1,2]. The main components of aircraft on which there is a high probability of ice accretion are the wings, tail, engine inlet, propeller blades, and sensors [3]. Airplane icing poses a great threat to flight safety. Throughout the history of domestic aviation, many aviation disasters have been caused by aircraft icing. Therefore, aircraft anti-icing technologies are getting attention from researchers.

To prevent the airplane icing phenomenon, experts have designed various aircraft deicing systems, including mechanical deicing, electro-thermal deicing, and liquid deicing [4,5]. Mechanical deicing mainly relies on acoustic waves, electricity, gas, or other physical media to generate mechanical forces on the ice surface of aircraft to destroy the ice structure and remove the ice [6]. Gas thermal deicing usually requires extra hot air emitted by the engine, which will reduce the efficiency of the aircraft. Electrothermal deicing electrifies the resistance element to heat up to prevent icing, which is often used to prevent icing of pitot tubes and windshields [7]. L. Vertuccio [8] et al. designed an electrically heated anti-ice system for aircraft surfaces using flexible graphene films, which are placed between two carbon fiber reinforced plastic (CFRP) laminates. Although the flexible heater has excellent mechanical and thermal performance, its electrothermal de-icing system is complex and consumes a lot of energy. Another common deicing method is to spray a liquid with a very low freezing point on a site that is likely to freeze, and mix it with supercooled water and freeze below the indicated temperature to prevent the icing phenomenon [9]. Y. Wang [10]

et al. used poly vinyl pyrrolidone in conjunction with poly acrylic acid as a thickener in water/glycol solutions for aircraft de-icing. Although this kind of deicing liquid has excellent deicing performance, its enormous utilization has adverse ecological effects [11].

With the recent developments of bionics, scholars have successfully imitated the surface microstructure of plants with their hydrophobic ability [12], inspired by the special wettability of the lotus leaf surface, and mastered the method of achieving a superhydrophobic surface with a certain roughness through low surface energy modification. It was found that the superhydrophobic surface has certain advantages in terms of delaying the surface droplet freezing and reducing the ice adhesion, which provides a new idea for aircraft anti-icing technology. The main methods of preparing anti-icing surfaces on metal are superhydrophobic coating, laser processing, and chemical etching. Renmei Dou [13] et al. prepared an anti-icing coating with an aqueous lubricating layer, which is formed by absorbing water or water from melted ice. This coating can be directly applied to various substrates, such as metals, metal alloys, ceramics, and polymers, and the ice adhesion strength of the coating surface can be reduced to 30 kPa. A breeze can blow off the ice formed on top of the coating in the wind tunnel, and its aqueous lubricating layer remains even at −53 °C. Zuojia Wang [14] et al. obtained a hydrophobic coating with a static water contact angle of 147° by using an aluminate coupling agent on an aluminum surface. Experimental results have shown that frost deposition on the hydrophobic coating was delayed within 60 min compared with the surface without coating. The anti-ice coating method is flexible, and could be applied to a variety of materials, but the coating has poor stability, a short action time, high maintenance cost, and is easy to scratch or damage. Ludmila B. Boinovich [15] et al. prepared an excellent anti-icing coating on an aluminum alloy surface based on the fine tuning of both laser processing and protocols of deposition of the fluoro silanes on nanotextured surfaces. The experimental results indicated that droplets deposited on the coating at −20 °C temperature can remain supercooled liquid without becoming frost crystals for up to 50 min, while at −10 °C, the droplets can remain liquid for more than 2000 min. Victor Rico [16] et al. processed the Al6061 material by laser, and further modified it with a nanostructured $Al_2O_3$ thin film so that its surface can reach the dual roughness and porous state. The laser processed Al6061 material was grafted with perfluorooctyltriethoxysilane (PFOTES) steam to make it superhydrophobic, which extended the freezing time of water droplets to nearly 4 h at −5 °C, and showed significant low ice deposition in wind tunnel tests. Wei Xing [17] et al. directly constructed the grating textures of hierarchical tertiary structures on the aluminum alloy surface using picosecond laser processing. These layered texture surfaces showed good hydrophobic properties at low temperatures. In comparison with the original surface of aluminum alloy, the freezing time of water droplets on the surface of the aluminum alloy after laser processing is delayed from 2879 s to 4015 s, and the freezing temperature is as low as −23 °C, showing excellent anti-icing performance. The key advantage of laser-machined superhydrophobic anti-ice structures is that the morphology of the prepared surface microstructure is highly flexible. Min Ruan [18] et al. simplified the preparation process of the superhydrophobic surface on an aluminum alloy substrate through electrochemical anodic oxidation and chemical etching methods, and conducted anti-icing experiments. Starting from room temperature (16.0 °C), the icing time of superhydrophobic surface can be delayed from 406 s to 676 s, compared with ordinary aluminum alloy surface. The icing temperature can be reduced from −2.2 °C to −6.1 °C. The results showed that the superhydrophobic surface prepared using this method has excellent ice resistance. Zhiping Zuo [19] et al. proposed a method to prepare a coral-like superhydrophobic structure on the surface of aluminum through chemical etching and hot water treatment to improve its ice resistance. The contact angle of the structure can reach 164.8 ± 1.1°, and the sliding angle is less than 1°. Water droplets on the surface can remain ice-free for more than 110 min, showing good anti-ice performance. Yang [20] et al., using hydrochloric acid etching at a certain concentration on the surface of aluminum alloy, successfully developed a simple method to prepare the superhydrophobic surface of an aluminum alloy. This superhydrophobic surface also

showed significant ice resistance, and the freezing time of droplets on the surface could be delayed to 600–700 s from $-5\,°C$ to $-8\,°C$. The preparation of superhydrophobic surface through chemical etching method is easy to achieve and takes a short time. However, the hydrophobic structures prepared by the above chemical etching methods typically only had a single-stage structure, thus the anti-icing effect was not as good as the hierarchical structure. The anti-icing surface preparation methods mentioned above had still some limitations, such as low process efficiency, complex process, and difficulty in the efficient preparation of anti-ice surfaces on a large area. However, solving these problems is of great significance to improve the safety performance of aircraft.

This paper proposes a hybrid laser machining–wet etching to prepare superhydrophobic micro/nanostructures on the aluminum alloy surface to realize anti-ice function. This method has numerous advantages, such as simple equipment, convenient operation, short process time, environmental protection, low cost, and ease of application. The anti-icing effect of the superhydrophobic surface on the aluminum alloy surface was studied by the icing delay test, ice adhesion experiment, and dynamic water repellency experiment.

## 2. Experiment

### 2.1. Reagents and Instruments

An aluminum alloy (2524) was selected as the substrate material. The technical parameters of the 2524 aluminum alloy are shown in Table 1. The equipment used includes a fiber laser marking machine (SK-CX20, Shanghai Sung Laser Co., Ltd., Shanghai, China), an electron microscope (Regulus8200, HITACHI, Janpan), a confocal microscope (OLS5000, Olympus, Janpan), and a contact angle measuring instrument (OCA25, DataPhysics Instruments, Filderstadt, Germany). Chemicals and reagents used in the experiment include anhydrous ethanol, acetone, hydrochloric acid, and the purity of 99% perfluorooctyl ethyl trimethoxysilane (FAS-17). The above reagents are made by the Sinopharm group.

**Table 1.** The technical parameters of 2524 aluminum alloy (wt%).

| Cu | Mg | Mn | Fe | Si | Ti | Al |
|---|---|---|---|---|---|---|
| 4.45 | 1.48 | 0.66 | 0.005 | 0.035 | 0.023 | Bal |

### 2.2. Experiments Processing

#### 2.2.1. Pre-Experiment Workpiece Preparation

The sample was machined into a 20 mm $\times$ 30 mm $\times$ 2 mm sample block by wire-cut electric discharge equipment. Before the experiment, the cut samples were polished with sandpaper in turn. After polishing, the samples were ultrasonically cleaned with acetone, alcohol, and distilled water for 10 min, then dried and stored with alcohol for future usage.

#### 2.2.2. Hierarchical Structure Processing

The microstructure (MS) was processed by laser engraving. The selected laser processing parameters had a scanning interval of 60 μm, laser power of 14 W, scanning speed of 100 mm/s, pulse frequency of 20 kHz, pulse width of 10 ns, spot diameter of 55 μm, wavelength of 1064 nm, and single pulse energy for $6 \times 10^{-4}$ J, and the number of laser passes was 10. The fluorinated microstructure's contact angle (CA) can reach 149.2° at room temperature ($25 \pm 1\,°C$). The nanostructure (NS) was prepared by wet etching, and the process sequence is shown in Figure 1. The reaction time was kept at 10 min, the reaction temperature was set at 30 °C, and the concentration of hydrochloric acid was selected as 3 M. Firstly, the sample was immersed in hydrochloric acid for reaction, and then it was dipped into dilute sodium bicarbonate solution to remove the residual acid solution on the sample surface. The samples were then taken out of the solution and underwent ultrasonic cleaning. After that, the samples were placed in a 0.8 wt% FAS-17 solution, soaked for 1 h, and heated in an oven at 140 °C for 1 h to obtain nanoscale superhydrophobic surface samples. After fluorinating, the CA of the nanostructure could reach 152.1° at room tem-

perature. The average surface roughness of NS was 1.17μm. Finally, laser machining and chemical etching obtained the hierarchical structure (H.S.). The process steps for preparing the hierarchical structure are shown in Figure 2. After preparing the layered structure, the sample was soaked in a 0.8 wt% FAS-17 solution for one hour, and then this was heated off in a 140 °C oven for one hour to obtain a hyper-hydrophobic surface sample. At the same time, the contact angle and rolling angle were tested. The CA of the sample reached 154.2°, the rolling angle was approximately 8° at room temperature, and it showed good hydrophobicity. The SEM of MS, NS, and HS are shown in Figure 3a, b, c, and d, respectively. The groove spacing of the laser processed surface is approximately 60 μm. There were remelted materials on both sides of the groove that depicted that the structure had excellent hydrophobic properties. After chemical etching, a large number of NS appeared on the sample surface, which led to its superhydrophobic properties. After laser processing and chemical etching, a nanoscale structure on the surface of the microgroove improved the contact angle of the hierarchical structure. It can be seen from Figure 3e,f that the depth of the groove on the HS surface was approximately 52 μm, and the width of the groove was approximately 28 μm.

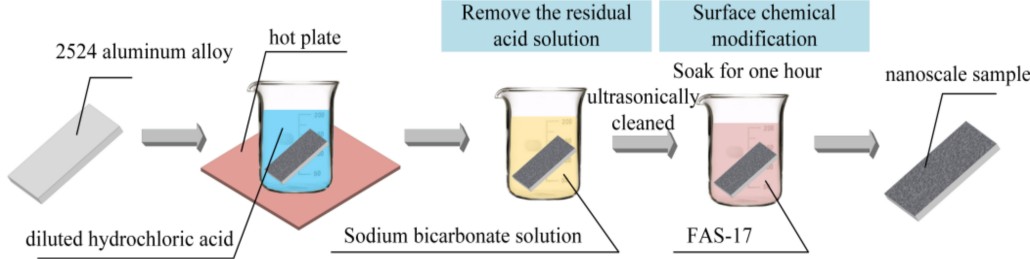

**Figure 1.** The wet etching procedure and key steps.

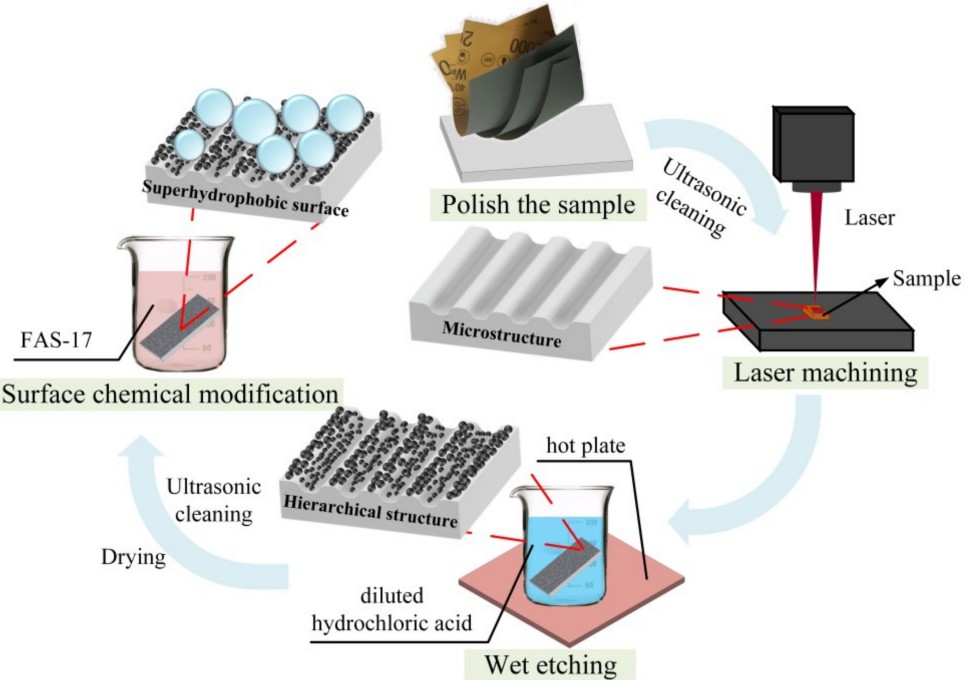

**Figure 2.** The overall preparation process of the hierarchical structure.

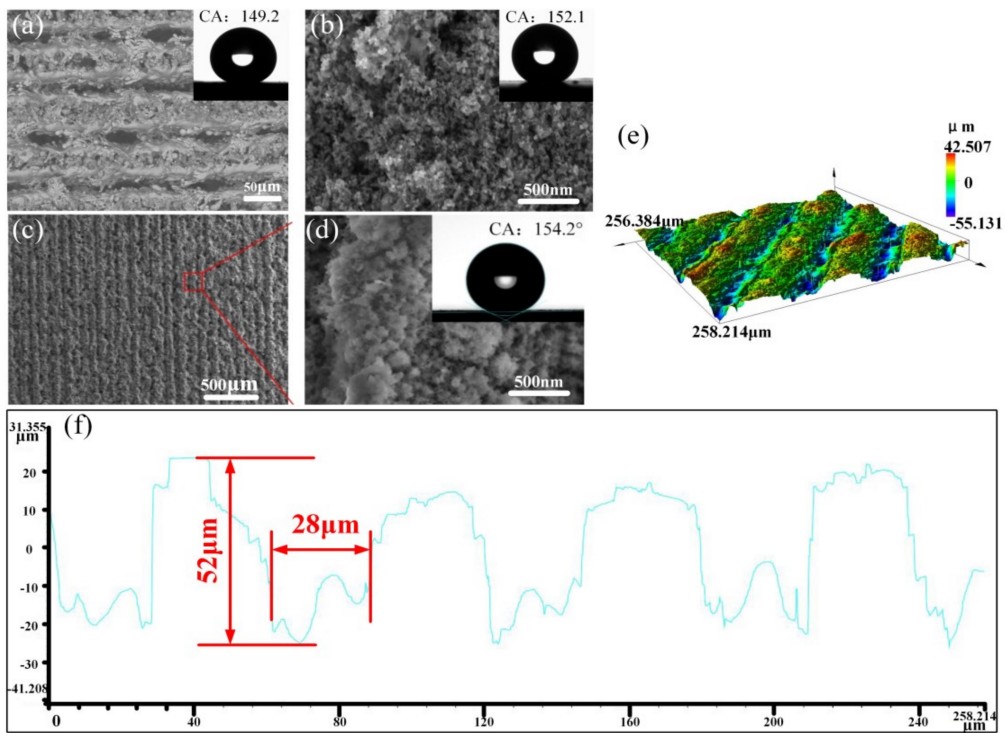

**Figure 3.** Surface topography of each structure. (**a**) The SEM of the microstructure; (**b**) the SEM of the nanostructure; (**c**,**d**) the SEM of hierarchical structure; (**e**) 3D morphology of the HS surface; (**f**) height profile of the HS surface.

### 2.2.3. Icing Delay, Ice Adhesion, and Dynamic Water Repellency Experiments

- Icing delay experiment:

Water droplets impact aircraft parts at various angles and in a wide range during flight, therefore it is necessary to consider the freezing time of droplets adhering to the surface. It is important to mention that prolonged icing time on the surface is beneficial for the aircraft to shake off liquid droplets during flight, and can be vaporized before the supercooled water droplets freeze through the auxiliary deicing system. This greatly relieves the pressure of the aircraft de-icing system. The icing delay experiment was carried out for this reason. The icing delay experiment was carried out on the smooth surface (SS), nanostructure surface (NS), microstructure surface (MS), and hierarchical structure (HS), the surfaces of which were all fluorinated. The icing delay test device is shown in Figure 4a. The anti-icing performance was evaluated by comparing the droplet freezing time on different surfaces at −10 °C. To carry out these experiments, three different sites were selected on material surfaces, and five values of icing delay time were recorded from each area, with the mean value taken. During the test, the relative humidity was 50~60%, and the dew point was about 13.9~16.7°.

- Ice adhesion experiment:

By delaying the freezing time of the droplets on the aircraft's surface, more time can be taken for dealing with droplets which are at risk of freezing. The strength of droplet adhesion on the surface determines the difficulty of dealing with these hidden dangers. Therefore, the adhesion strength of the borneol surface is very important for anti-icing and de-icing. The ice adhesion experiments were set to test the adhesion ability of different sample surfaces. The samples used in the experiment include SS, MS, NS, and HS. Initially, ice cubes of 10 mm × 10 mm × 10 mm were prepared and kept together with the sample in a refrigerator at a certain temperature for 24 h to obtain cubes of ice that adhered to the surface of the samples. Later on, the push–pull digital dynamometer was used to push the ice onto the surface of the sample. The computer was used to monitor the absolute value

in real-time, and plotted to record the peak values, and took the average value. The icing adhesion test device is shown in Figure 4b.

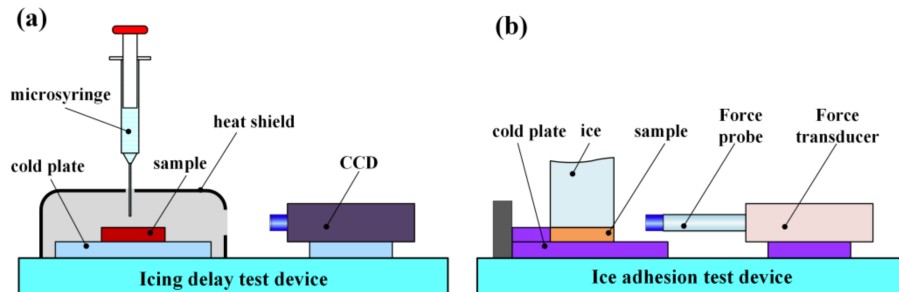

**Figure 4.** Schematic diagram of experimental device. (**a**) Icing delay test device; (**b**) icing adhesion test device.

- Dynamic water repellency experiment:

The ice formed when supercooled water droplets hit the surface of an aircraft and freeze is called frost ice, commonly forming at low temperatures (0 °C to −40 °C). Due to the small size of the impacted droplet, the freezing time is short, and the near-spherical droplets do not form a film of water on the aircraft's surface, but freeze directly in that shape. The type of frost ice is relatively regular, from the cross-section of a wedge shape, thus it is also called wedge ice. Although this type of icing has a small range and fragile texture, it still cannot be ignored, since it has a large impact on aircraft. The high-speed camera of the contact angle measuring instrument was used to record the dynamic contact process of a droplet hitting the sample surface, as well as observe droplets bouncing on different surfaces. The samples used in the experiment include SS, MS, NS, and HS. The droplet volume was 5 µL, the release height was 5 mm, the release speed was 5 µL/s, and the frame number of the high-speed camera was 350 FPS.

## 3. Results and Discussion

### 3.1. Icing Delay Experiment

By comparing the freezing times of the surfaces at −10 °C, it was found that untreated surfaces froze in a very short time, while treated surfaces increased this time by approximately 10 times. The delay in the crystallization of droplets when in contact the superhydrophobic surface is due to the elevated barrier of the supercooled water/ice phase transition [15]. Figure 5 shows the frozen state of four samples placed on the center of the temperature control platform recorded by a CCD camera. Figure 5a shows the average freezing time of different samples at −10 °C, and the standard deviation of HS, NS, MS, and SS freezing times are 59.44, 58.11, 56.07, and 11.57, respectively. It can be seen from Figure 5b,c that no small droplets appeared on the surface of MS and SS during the freezing process. The droplet added to the SS surface froze almost completely at about 175 s, except for the central region, and froze completely at 245 s. In addition, it can be seen from real shot pictures that many ice crystals appeared around the droplets on SS and MS when frozen. The droplet added to the MS surface froze completely at 2375 s. At the same time, the HS surface was slightly better than the MS in terms of time, with a delay of approximately 400 s. Preliminary analysis suggests that the microstructure had larger voids than the nanostructure. When the small droplets infiltrated the void, the actual contact area between the droplets and the low-temperature substrate increased, increasing the heat transfer rate and resulting in a faster heat conversion. The comparison of nanostructure and the hierarchical structure demonstrated little difference in terms of freezing time, but there was a big difference in the ice crystal spreading. As shown from Figure 5d,e, both N.S. and H.S. surfaces were covered with small water droplets at approximately 1200 s. There are parts of the NS surface that froze ahead of the water droplet. The reason may be because the NS sample was not etched with a consistent velocity in the hydrochloric

acid for the whole surface, due to the uneven grooves caused by sandpaper grinding. This phenomenon would result in an ununiform morphology or even an unetched consequence of some areas, which would have had further effects on the overall anti-ice performance of the sample. However, it can be observed that the surface ice of HS was gradually covered from the periphery to the center, with no area frozen in advance.

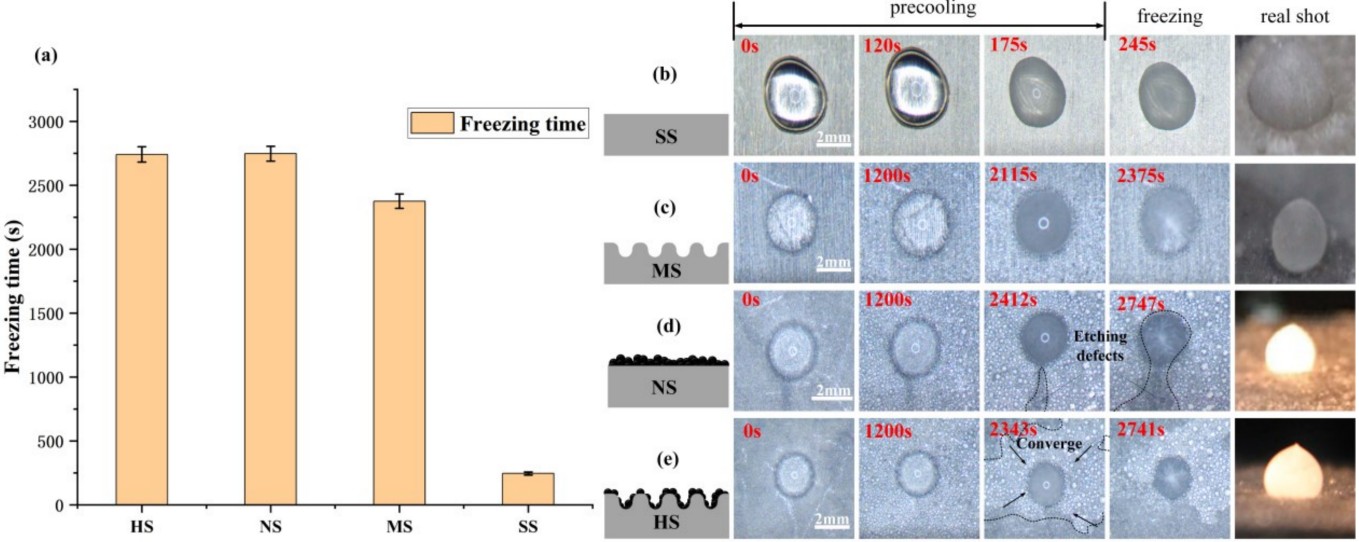

**Figure 5.** Ice conditions of four samples, (**a**) Ice conditions of SS; (**b**) ice conditions of MS; (**c**) ice conditions of NS; (**d**) ice conditions of HS; (**e**) histogram of freezing time of four samples.

### 3.2. Ice Adhesion Experiments Analysis

These experiments presented the ice adhesion strength on four different fabricated structures. Figure 6a shows the real-time curve record of the force monitored by the dynamometer, and its peak value was recorded several times and averaged.

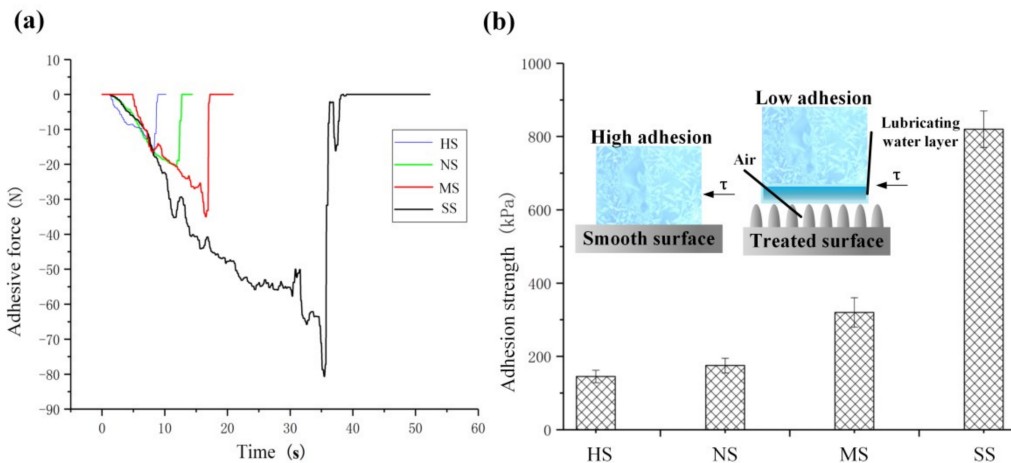

**Figure 6.** Adhesion test results (**a**) Monitoring curve of dynamometer; (**b**) ice adhesion strength.

It can be seen from Figure 6b that, in the SS sample, the adhesive force of the ice layer was approximately 800 kPa, while 200 kPa in the NS sample, and 170 kPa in the HS sample, which was 75% lower than that in the untreated sample. The triple point of water at the ice/gas and ice/solid surface contact interface migrated to the lower part and increased the temperature. This resulted in a water layer with negative temperature between the ice and the sample surface contact interface [21,22]. The water layer at the contact interface between ice and sample acted as a lubricating layer, significantly reducing the adhesion strength

of ice under a shear load. The reason may be that relative movement occurs between the inner layers of the lubrication layer under shear action, and static friction force does not need to be overcome. This would require less energy to remove surface ice, therefore less adhesive. Thus, in some cases, spontaneous deicing may occur due to a combination of gravity, wind, vibration, or airflow pressure. On the other hand, due to a lubricating water layer at the contact interface, the surface modified with low surface energy cannot form covalent chemical bonds with the water-based media components, which further reduces the bonding strength between the ice and the treated sample.

### 3.3. Dynamic Water Repellency Experiment

Figure 7 shows the process of a droplet impacting the surface of four different samples. After impact, the droplet moved laterally until it was fully unfolded, at which point the radius was at its maximum. Then, the droplets gradually contracted and elongated vertically, and finally bounced and separated. The process can be summarized as impact–contact–expansion–contraction–bounce. The two parts from impact to expansion were determined by the volume and release velocity of the droplet, while the expansion part was mainly determined by the hydrophobic properties of the sample. Figure 7a,b show that both HS and NS can bounce after impact, indicating that the contact time between the droplets and the sample was short. The droplet touched the sample surface at 3 ms, bounced at 12 ms, and the dynamic contact time was less than 9 ms. However, the droplets did not bounce off the surface of MS and SS, as shown in Figure 7c,d. Starting from the 9 ms, MS repeated two cycles of spread and contraction, but still did not bounce and instead adhered to the sample's surface. The droplet directly adhered to the SS surface after colliding with it.

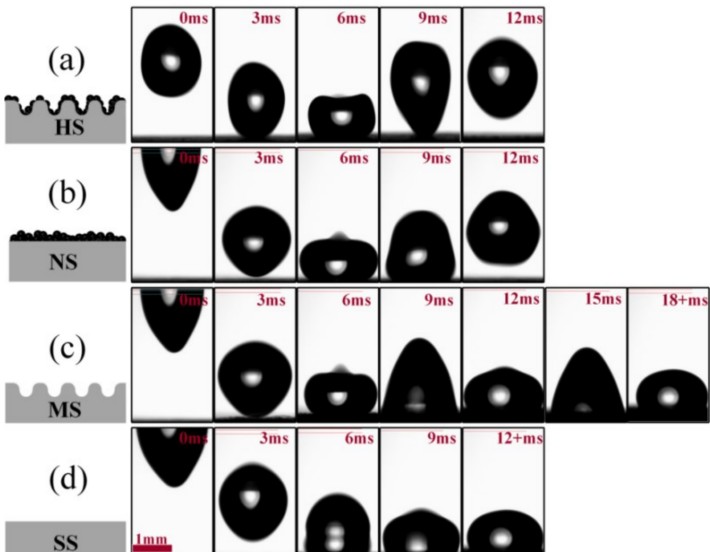

**Figure 7.** Schematic diagram of droplet bouncing process on the sample surface (**a**) droplets on the surface of HS; (**b**) droplets on the surface of NS; (**c**) droplets on the surface of MS; (**d**) droplets on the surface of SS.

To understand the relationship between the surface wettability and the solid–liquid contact time of droplet impact, the contact angles of four samples were measured. The contact angle hysteresis (the difference between forward contact angle and receding contact angle) of HS and NS samples were 5° and 7°, respectively, showing good dynamic hydrophobicity. The contact angle hysteresis of MS and SS samples were 48° and 55°, respectively. The MS sample had good static hydrophobicity (contact angle 142°), but poor dynamic hydrophobicity. Figure 8 shows the static contact angles, advanced contact angle, and receding contact angle of all samples.

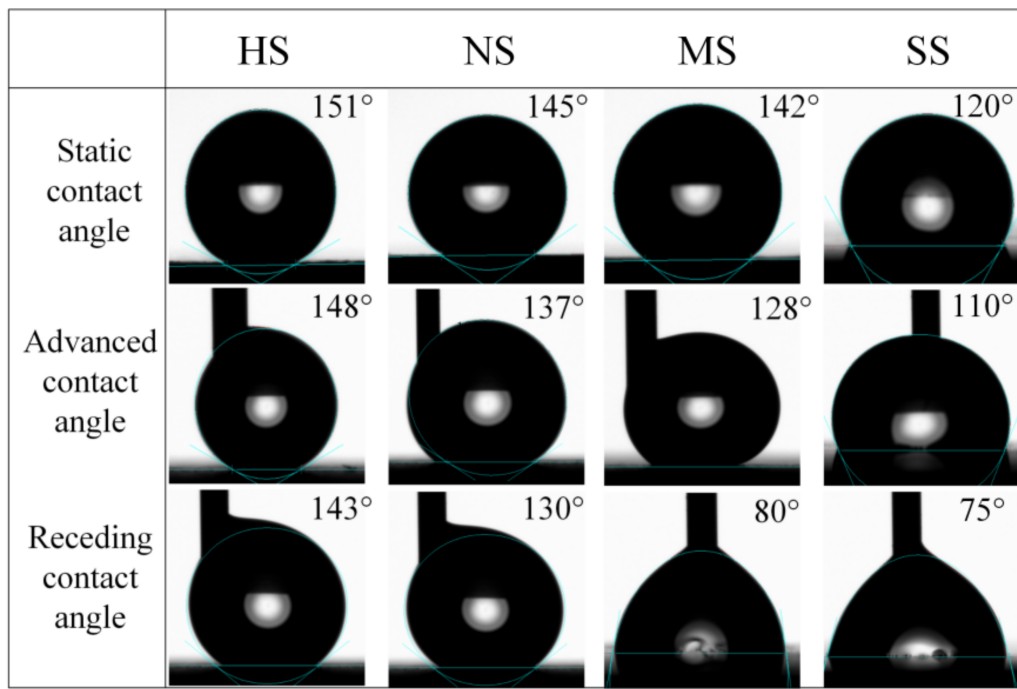

**Figure 8.** Measurement results of contact angle of four samples.

The above phenomenon indicates that the droplet that can bounce on the sample's surface is mainly determined by the size of the dynamic contact angle and the hysteresis of the contact angle. In contrast, the static contact angle enables the droplet to go through multiple cycles of spreading and contracting, and cannot directly affect the contact time between the droplet and the sample. Figure 9 shows the relationship between droplet contact time and sample surface wettability. HS and NS samples had the shortest solid–liquid contact time. In addition, the static hydrophobicity (HS had a static contact angle of 151°, and NS 145°) and dynamic hydrophobicity (hysteresis of contact angle HS: 5°, NS: 7°) were the best. The liquid droplets on samples MS and SS had an order of magnitude of solid–liquid contact time, and both adhered to the surface without bouncing.

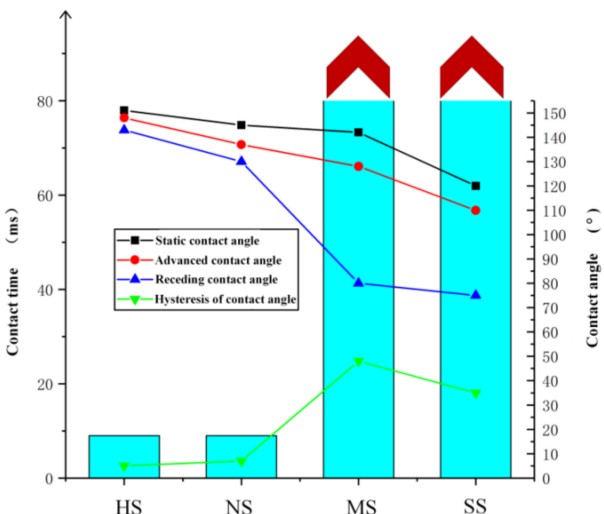

**Figure 9.** Relationship between contact angle and impact contact time of samples.

Meanwhile, MS had better static hydrophobicity and poor dynamic hydrophobicity, while SS had poor static and dynamic hydrophobicity. As the static contact angle decreased, the contact time of the droplet impact sample increased. Therefore, the contact

time of the droplet impact on the sample surface was not positively correlated with the static contact angle, but its trend was consistent with the hysteresis of contact angle. It indicated that the contact time of droplet impinging on the sample surface was mainly determined by dynamic wettability, such as the hysteresis of contact angle, but not by the static contact angle.

As the droplets hit the surface of the HS sample at a certain speed, the droplets could not exclude the air captured inside the MS and retain the Cassie wetting state. Therefore, the air captured during the droplet rebound can offset the external atmospheric pressure, thus reducing the viscous resistance of the droplet on the sample surface, resulting in the droplet being able to bounce in a short time. The droplets on the untreated surface become completely in contact with the sample surface, and the kinetic energy is consumed by the resistance to overcome the viscous resistance from the solid surface, and finally cannot bounce. Figure 10 shows the contact mechanism of the droplet spreading and contracting process when impacting SS and HS interfaces.

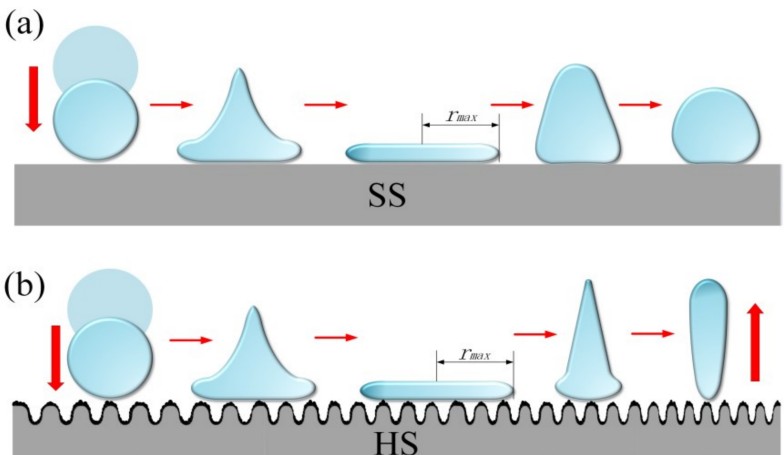

**Figure 10.** Droplet impact process. (**a**) Droplet impact process on SS; (**b**) droplet impact process on HS.

The above process can be explained by theoretical analysis. It is assumed that the droplet was subjected to continuous and uniform force throughout the process of contact–deploy–contraction. In the expansion stage [23], the relation between the ratio of the maximum spread radius to the initial radius and the contact angle is as follows:

$$\frac{D_{max}}{D_0} = \sqrt{\frac{W_e + 12}{3(1 - cos\theta) + 4\frac{W_e}{\sqrt{R_e}}}} \tag{1}$$

where $D_{max}$ is the maximum expansion radius after the droplet impacts the surface, $D_0$ is the initial falling radius of the droplet, and $W_e$ is the Weber number, representing the ratio of the kinetic energy of the droplet to the surface energy. '$R_e$' is the Reynolds number, representing the ratio of the inertial force to the viscous force of the droplet:

$$W_e = \frac{\rho D_0 V^2}{\sigma} \tag{2}$$

$$R_e = \frac{\rho D_0 V}{\mu} \tag{3}$$

where $\rho$ is the droplet density (1 kg/m$^3$), $\sigma$ is the liquid–gas interfacial tension, $V$ is the impact velocity, and $\mu$ is the liquid viscosity.

As per the above formulas, as the droplet impacted the sample surface, the larger the contact angle was, the more the expansion radius decreased. Since the initial kinetic

energy and potential energy were the same when the droplet was released, part of the energy was lost during the contact between the droplet and the surface and the process of the droplet jumping up. According to Newton's laws, droplet adhesion to a non-ideal solid surface is caused by surface resistance. Different results are obtained when the same droplet impinges on different samples with the same initial kinetic energy and potential energy, indicating that the resistance described in the law is the main reason for different results. The adhesion resistance *F* is the resultant force of the projection of droplet surface tension in the expansion direction. The work performed to overcome resistance is as follows:

$$W = \oint Frdr \tag{4}$$

It can be seen from Equation (4) that *W* is directly proportional to the droplet developing radius. The smaller the radius, the smaller the value of *W*. Combined with the above conclusions, it can be shown that the larger the contact angle, the smaller the value of *W*. According to energy conservation, the more energy left for the bounce, the shorter the solid–liquid contact time. To explain the solid–liquid contact time from the energy perspective, it is assumed that the initial droplet is a sphere, no droplet breakage occurs in the collision process, and that it is cylindrical at its maximum spread. The droplet release rate is approximately 0. Since the drop height is small, the resistance in the drop process is also approximately 0. Then, kinetic energy $E_{K1}$, surface energy $E_{S1}$, and potential energy $E_{H1}$ before impact are respectively:

$$E_{K1} = \frac{\pi \rho D_0{}^3 V^2}{12} \tag{5}$$

$$E_{S1} = \pi D_0{}^2 \sigma \tag{6}$$

$$E_{H1} = \frac{\pi \rho g D_0{}^4}{12} \tag{7}$$

When the droplet spreads to the maximum, the kinetic energy is 0, and the surface energy and potential energy are expressed as:

$$E_{S2} = \frac{\pi D_{max}{}^2 (1 - cos\theta^*)\sigma}{4} \tag{8}$$

$$E_{H2} = \frac{\pi \rho g D_{max}{}^2 \delta^2}{8} \tag{9}$$

where $\delta$ is the thickness when the droplet is spread to the maximum extent. When the droplet spreads to the maximum, the work completed to overcome the adhesion force is converted into the form expressed in time:

$$W = \int_0^t \int_\Omega \Psi d\Omega dt \tag{10}$$

where droplet volume $\Omega$, viscous and dissipation function $\Psi$ and $\delta$ are respectively expressed as:

$$\Omega = \frac{\pi \delta D_{max}{}^2}{4} \tag{11}$$

$$\Psi = \mu \left(\frac{V}{\delta}\right)^2 \tag{12}$$

$$\delta = 2\sqrt{\frac{\mu D_0}{\rho V_0}} \tag{13}$$

Substituting the above into Equation (10), then:

$$W = \frac{\pi V^2 D_{max}^2}{8} \sqrt{\frac{\rho \mu V}{D_0}} t \tag{14}$$

According to the law of energy conservation, time $t$ can be calculated as:

$$t = \frac{E_{K1} + E_{H1} + E_{S1} - E_{H2} - E_{S2}}{\Psi \Omega} \tag{15}$$

Combined with the above formulas, the droplet can be obtained from contact to spread to the maximum time $t$. The droplet will shrink after spreading to the maximum, and the solid–liquid contact time should be the sum of spreading and shrinking time. The liquid film is formed at the edge of the droplet during the process of shrinkage, and the inertia force and surface tension at the edge greatly influence the process of shrinkage. The momentum relation at the edge of the liquid film [24] can be written as:

$$\frac{d}{dt}\left(m\frac{dr(t)}{dt}\right) = F_c \tag{16}$$

where $m$ is the liquid film edge mass, $r$ is the liquid film radius, and $F_c$ is the liquid film capillary force. The capillary force of the liquid film is approximate:

$$F_c \approx 2\pi r(1 - cos\theta_R)\gamma \tag{17}$$

The change rate of the liquid film quality with time is:

$$\dot{m}(t) = 2\pi \rho r(t) V_{ret} \delta \tag{18}$$

The droplet shrinkage speed is expressed as $V_{ret} \approx \sqrt{\gamma(1 - cos\theta/\rho\delta)}$, and the droplet thickness is expressed as $\delta = \frac{2D_0^3}{3D_{max}^2}$ according to the conservation of mass. Then, the velocity relationship is $V_{ret} \sim D_{max}\sqrt{1 - cos\theta}(\rho D_0^3/\gamma)^{-1/2}$, and the magnitude of shrinkage time is:

$$t_{ret} \sim \sqrt{\rho D_0^3/\gamma} \tag{19}$$

According to the above formulas, the magnitude of the liquid–solid contact time of the droplet on the HS sample can be calculated. The droplet diameter is set as 2 mm, the surface tension is set as 72 MN/m, and the Weber number and Reynolds number are approximately 28 and 2000, respectively. Finally, the total contact time is 10 ms, consistent with the experimental result of 9 ms.

## 4. Conclusions

This research work proposes a hybrid laser processing technology and wet etching technology to prepare a hierarchical structure on a 2524 aluminum alloy. In addition, an icing delay experiment, ice adhesion experiment, and dynamic water repellency experiment were carried out to verify the excellent anti-icing performance of the sample to improve aircraft safety. Based on the experimental results and theoretical analysis, the key conclusions are summarized as follows:

(1) Specific laser processing parameters and wet etching parameters can effectively prepare a uniform hierarchical structure on the surface of the aluminum alloy. The structure can meet the standard of super-hydrophobicity.

(2) The freezing time of droplets on the treated sample surface was approximately 10 times longer than that of the untreated sample. The nanostructure samples can maintain good hydrophobicity after the droplet is frozen and melted, which means that it has a certain degree of durability against freezing.

(3) The ice adhesion experiment on the surface of the sample shows that the adhesion strength of the treated sample decreased by approximately 75% compared to the untreated samples. This indicates that the ice can easily be removed even when the surface is frozen.

(4) The dynamic water repellency experiment revealed that the contact time between the impinging droplet and sample surface mainly depends on the dynamic wettability of the sample surface, that is, the hysteresis of contact angle. The smaller the hysteresis of the contact angle, the shorter the contact time between the impinging droplet and the sample surface. The contact time between the droplet and sample obtained by the experiment is consistent with the theoretical calculation.

**Author Contributions:** X.H. and A.X. were responsible for the overall planning of the paper, L.H. was responsible for the establishment of the icing adhesion test device and icing delay test device, A.X. carried out the icing delay experiment, S.Q. and J.Z. carried out the ice adhesion experiment and dynamic water repellency experiment, H.L. and N.H. guided the completion of data processing and paper writing. All authors have read and agreed to the published version of the manuscript.

**Funding:** This research was funded by [the National Natural Science Foundation of China] grant number [No. 51875285], [the Natural Science Foundation of Jiangsu Province] grant number [No. BK20190066], [College Young Teachers Fund of the Fok Ying Tung Education Foundation] grant number [No. 20193218210002], and [the Fundamental Research Funds for the Central Universities] grant number [NO. NE2020005].

**Institutional Review Board Statement:** Not applicable.

**Informed Consent Statement:** Not applicable.

**Data Availability Statement:** Not applicable.

**Acknowledgments:** The authors sincerely thank Muhammad Jamil and Qaisar Ali of Nanjing University of Aeronautics and Astronautics (NUAA) for helping, editing, and polishing the language of this manuscript.

**Conflicts of Interest:** The authors declare no conflict of interest.

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
