# Peer review of "Fabrication of an Anti-Icing Aluminum Alloy Surface by Combining Wet Etching and Laser Machining"

_applsci, doi:10.3390/app12042119_

Round 1

Reviewer 1 Report

This is an interesting work on anti-icing surfaces through facile micro/nano scale modifications.   The work is up-to-date and has a few potential applications in the field. 

It can be accepted for publication subject to some revision, with major ones listed below:

  1. The surface morphology is maybe one of the most important part of this paper, however there are very few details about this:
    1. Only top-view SEM images were included, any “cross-section” views?
    2. Are there any “dual roughness” measurement results?
    3. Are there any 3D information of the hierarchy surface structures?
  2. There are many grammar mistakes, confusion, and lack of clarities. A few examples below:
    1. In the abstract, “fabricated with combined these two kinds of processing”
    2. In the abstract, “can be delayed by ten times”

And many more in the main context – a thorough review and improvement is required.

There are also a few minor corrections needed:

  1. More clarification is required regarding the dynamic water repellent experiment:

For example, what’s the relationship between droplet size, release speed, release height and impact velocity? 

  1. More clarification is required when discussing “ice crystal spreading” – more information needs to be added to Figure 5 and relevant paragraphs.

  1. There needs to be more references added to support discussions in the Introduction part, for example:

The electrothermal deicing has a complex system and consumes large power.

A common practice for liquid deicing is to spray a liquid with a very low freezing point on a site that is likely to freeze, and to mix it with supercooled water and freeze below the indicated temperature to prevent it from freezing.

The effective time of liquid deicing is usually short, and it will pollute the environment.

 etc...

Reviewer 2 Report

  1. The English needs to be improved. For example, in abstract "facile" in inappropriate.
  2. Typos need to be corrected. For example, in section 1.2.2 "scanning speed for 100 mm/s 15".
  3. Were the samples polished by hand with sandpaper? Will the difference in the workpiece surface affect the result?
  4. What are the microstructure dimensions? Justify them with experiment or reference.
  5. Fig 3 (a) & (c) lack scale.

Reviewer 3 Report

The manuscript is devoted to the fabrication of composite superhydrophobic coatings on aluminum alloy surface. The manuscript cannot be recommended for publication due to numerous flaws. Some of these flaws are listed below.

  1. It is difficult to read the manuscript because the lingual errors such as wording, grammar, and style do not allow the reader to get access to the scientific content of the manuscript and it is recommended first to correct all the text.
  2. The introduction is poorly written and does not rationalize the choice of the procedures for fabrication of the hierarchical superhydrophobic surface. Usually, the nanosecond laser treatment already creates hierarchical structure suitable for obtaining the superhydrophobicity after surface modification with low-surface-energy substance (hydrophobic agent). The chemical etching with acid makes the surface more fragile, which is harmful for surface durability in icing conditions.
  3. The experimental part is poorly described. The procedure of laser treatment lacks many important details like pulse duration, focused beam size, pulse energy or effective laser fluence etc.
  4. The discussion of the results obtained in the paper is at scientifically low level, the assertions are very naïve and, in many cases, wrong. Only a couple of examples (the list can be easily continued):

4a) Why "The adhesion strength of borneol surface is very important"? (line 170)

4b) "When the droplet melted, the small ice particles can easily enter into the micron level gap..." (lines 246-247).  Why? The ice is lighter than water, and small ice particles should flow upward, not downward "into the micron level gap"!

  1. The choice of experimental tests is not suitable to really probe (prove) the anti-icing ability of developed coatings, the experimental conditions (cool Peltier stage in the warm laboratory environment) are poorly defined and poorly reproducible from test to test, no statistical data were provided for “freezing delay” (which should not include the cooling time of the droplet to the target temperature, only the time between the instant when the droplet reached the target temperature to the instant of droplet freezing, see, for example, doi: 10.1021/acsnano.8b09549)
  2. The data obtained in this study should be compared against other anti-icing coatings for aluminum alloys reported in the recent literature.

Round 2

Reviewer 1 Report

Figure quality is still poor in some cases (e.g. figure 3) - please improve. 

Reviewer 3 Report

The revision of the manuscript was not satisfactory. The English still needs considerable revision, not just a polishing of grammar. Some examples of poor style:

“recorded the icing state by recording the icing state”

“SS adheres directly to the surface of the sample after contact”

“Compared with untreated samples, treated samples can prolong the freezing time by about 10 times, which the icing delay experiment can see”

“When the ice layer was formed in a large sample area, the droplet would freeze, indicating that the uniformity of the sample's surface with HS was good” Why? How the freezing event in the droplet is related to the uniformity of the surface?

The experimental procedures are still unclear, and the description lacks the statistical analysis of the data. How many droplets were used to determine the freezing delay time? What were the average values and standard deviations? There are also some discrepancies in the description of icing delay experiment. In lines 181-182: “Keep the temperature at -10 °C;”, but in line 185: “Controlled the temperature at -20 °C”

It is difficult to be convinced that “The surface roughness parameters Ra of … HS surface is 5.54 μm” (lines 152-153) given that in Figure 3f, the height profile shows depth of the groove about 52 μm, and the width of the groove about 28 μm.

On the comments 5-6 of the previous report (concerning the comparison of their results on freezing delay with the recent literature data), the authors replied, that there are some data (“Yang et al.”, who obtained delay of 600-700 s at -5 °C-8 °C) which are inferior to their study. However, why did not they compare to better examples like paper mentioned by this Reviewer (doi: 10.1021/acsnano.8b09549), where the freezing delay at -15 °C extended over tens of hours?
